# A School-Based Randomized Controlled Trial to Promote Cycling to School in Adolescents: The PACO Study

**DOI:** 10.3390/ijerph18042066

**Published:** 2021-02-20

**Authors:** Palma Chillón, Patricia Gálvez-Fernández, Francisco Javier Huertas-Delgado, Manuel Herrador-Colmenero, Yaira Barranco-Ruiz, Emilio Villa-González, María Jesús Aranda-Balboa, Romina Gisele Saucedo-Araujo, Pablo Campos-Garzón, Daniel Molina-Soberanes, José Manuel Segura-Díaz, Fernando Rodríguez-Rodríguez, Amador Jesús Lara-Sánchez, Ana Queralt, Javier Molina-García, Enrique García Bengoechea, Sandra Mandic

**Affiliations:** 1PROFITH “PROmoting FITness and Health through Physical Activity” Research Group, Sport and Health University Research Institute (iMUDS), Department of Physical Education and Sports, Faculty of Sport Sciences, University of Granada, 18011 Granada, Spain; pchillon@ugr.es (P.C.); mhc@ugr.es (M.H.-C.); mjab@ugr.es (M.J.A.-B.); rgs@ugr.es (R.G.S.-A.); pcampos@ugr.es (P.C.-G.); jmsegdia@ugr.es (J.M.S.-D.); 2“La Inmaculada” Teacher Training Centre, University of Granada, 18013 Granada, Spain; fjhuertas@ugr.es; 3PROFITH “PROmoting FITness and Health through Physical Activity” Research Group, Sport and Health University Research Institute (iMUDS), Department of Physical Education and Sports, Faculty of Education and Sport Sciences, University of Granada, 52005 Melilla, Spain; ybarranco@ugr.es (Y.B.-R.); evilla@ugr.es (E.V.-G.); 4Department of Preventive Medicine and Public Health, Faculty of Medicine, University of Granada, 18016 Granada, Spain; danielmolinamd@gmail.com; 5IRyS Group, School of Physical Education, Pontificia Universidad Católica de Valparaíso, 2374631 Valparaíso, Chile; fernando.rodriguez@pucv.cl; 6Department of Didactics of Musical, Plastic and Corporal Expression, University of Jaén, 23071 Jaén, Spain; alara@ujaen.es; 7AFIPS Research Group, University of Valencia, 46010 Valencia, Spain; ana.queralt@uv.es (A.Q.); Javier.molina@uv.es (J.M.-G.); 8Department of Nursing, University of Valencia, 46010 Valencia, Spain; 9Department of Teaching of Musical, Visual and Corporal Expression, University of Valencia, 46022 Valencia, Spain; 10Physical Activity for Health, Health Research Institute, Department of Physical Education & Sport Sciences, University of Limerick, V94 T9PX Limerick, Ireland; enrique.garcia@ul.ie; 11School of Sport and Recreation, Faculty of Health and Environmental Sciences, Auckland University of Technology, Private Bag 92006, 1142 Auckland, New Zealand; sandy.mandic@aut.ac.nz

**Keywords:** active transport, youth, bicycle, exercise, school intervention

## Abstract

This manuscript describes the rationale and protocol of a school-based randomized controlled trial called “Cycling and Walk to School” (PACO, by its Spanish acronym) that aims to promote cycling to and from school and physical activity (PA) in adolescents. This study will examine the effects of this intervention in cycling and active commuting to and from school (ACS), PA and several ACS-related factors based on self-determination theory (SDT) and a social-ecological model (SEM). A total of 360 adolescents attending six high schools (three experimental and three control) from three Spanish cities will participate in this randomized controlled trial. The intervention (four cycling sessions; 1–2 h per session, one session per week) will be conducted by the research staff; the control group will continue their usual activities. PA levels will be measured by accelerometers, whereas ACS and the other study variables will be self-reported using questionnaires at baseline and post-intervention. The primary outcomes will be: rates of cycling to school, ACS and PA levels. In addition, SDT-related variables and individual, interpersonal, community, and environment variables relevant to ACS will be based on SEM. The findings will provide a comprehensive understanding of the short-term effects of this school-based intervention on cycling to school behaviour, ACS and PA levels in Spanish adolescents.

## 1. Introduction

### 1.1. Active Commuting to School: Benefits, Prevalence and School-Based Interventions

The physical, psychological, and social benefits of meeting physical activity (PA) recommendations (i.e., 60 min of daily moderate-to-vigorous PA [MVPA]) have been well-documented in the scientific literature [1]. However, four out of five adolescents do not meet these MVPA guidelines [2]. Specifically, in Spain, 76.6% of adolescents between 11 to 17 years old are insufficiently active [2]. Since previous studies have shown that PA patterns in childhood track to adulthood [3], promoting regular PA levels in young people is a public health concern [4].

Active commuting to and from school (ACS) is defined as any walking or cycling to and from school; while passive commuting refers to the use of motorized vehicles as a mode of commuting, such as by car, bus, metro, train, or motorcycle, among others [5]. ACS in adolescents is a routine behaviour that is associated with higher levels of PA on schooldays [6,7,8], less time spent in sedentary pursuits [9] and has social and mental benefits [10]. Compared to walking, cycling to school leads to significantly higher physiological benefits, particularly regarding cardiorespiratory fitness, due to its higher PA intensity [6]. In addition, ACS also has environment and economic benefits [11]. Moreover, getting young people physically active during their trip between home and school is currently one of the main priorities of global policies, such as the Global Action Plan on Physical Activity 2018–2030, promoted by the World Health Organization [12], and the 2030 Agenda for Sustainable Development, promoted by the United Nations [13].

However, despite the well-known benefits associated with ACS, fewer than half of adolescents globally use ACS [14], with large variability across countries [9]. In Spain, about 50% of adolescents reported walking to school and only 0.3% reported cycling to school [15]. Although 68.2% of Spanish adolescents and young people aged 12 to 24 years already own a bicycle and 92.2% of them could ride a bicycle [16], less than 2% reported cycling to school [15,17,18,19]. Given that cycling to school tends to track from childhood to adolescence [20], along with its documented benefits, promoting this health-related behaviour should be a public health priority.

Schools have been identified as an ideal setting to promote health-related behaviours such as ACS [21]. However, previous systematic reviews about school-based ACS interventions found non-significant results or small effect sizes [22,23,24,25]. These studies did not differentiate the effects on both walking and cycling to school because active commuting modes were pooled. Future school-based interventions should focus on improving walking or cycling to school separately because these two active modes of commuting have been associated with different health benefits, correlates, and require different skills and equipment [26,27]. However, most ACS interventions to date focused on walking to school (25 studies among 37), whereas studies that examined cycling interventions largely focused on primary school children (7 studies among 12) [22,25]. Therefore, there is a need to evaluate the effects of cycling interventions in secondary schools [23], particularly since the correlates of cycling to school between children and adolescents may also differ [28]. A recent scoping review [29] conducted on cycle skill training interventions reported than three of these six identified studies increased the cycling rates to school [28,30] and most of them reported an increase in student’s general cycling confidence, knowledge, skills, and attitudes [29]. School-based studies to promote ACS conducted to date have had several limitations including lack of use of theories of behaviour change, weak methodological design, and lack of reliable and valid assessment [22,23,24,25,29]. Given that most of the previous school-based cycling interventions have not been based on behaviour change theory, and/or have not used comprehensive and rigorous frameworks to assist with design, implementation, and evaluation of the interventions, further cycle skills training interventions are required to examine whether well-designed interventions can be effective in improving cycling to school [31].

### 1.2. Behaviour Change and Implementation Frameworks: Social-Ecological Model, Self-Determination Theory, and Reach Effectiveness Adoption Implementation Maintenance (RE-AIM) Framework

Findings from promising school-based ACS interventions suggest that the integration of behaviour change frameworks, such as the social-ecological model (SEM) [32] and self-determination theory (SDT) [33], in the design, implementation, and evaluation of these interventions may enhance initiation and long-term maintenance of ACS behaviours [34,35,36,37,38]. The SEM considers that ACS is influenced by different factors: individual (e.g., sex, age, ethnicity, etc.), interpersonal (e.g., family and significant other support), community (e.g., social capital), built environment (e.g., distance, safety, walkability, etc.), and policy (e.g., school policies) [35,36,39,40]. Complementary to the SEM, SDT posits that interpersonal and individual factors can influence the motivational process for ACS [41]. According to SDT, a student’s perception of social support for ACS from others (e.g., parents, teachers, peers, etc.) may have an influence on the student’s basic psychological need satisfaction (i.e., autonomy, competence, and relatedness satisfaction in ACS) which, in turn, may influence autonomous motivation and health-related behaviours such as ACS. The integration of both frameworks provides a better understanding of the motivational processes to actively commute to school [41].

Considering both internal and external validity when evaluating the potential for public health impact of behaviour change interventions remains a priority [42]. The Reach Effectiveness Adoption Implementation Maintenance (RE-AIM) framework has been widely used for this purpose [42,43]. The RE-AIM framework includes reach (R), effectiveness (E), adoption (A), implementation (I), and maintenance (M) [42]. Effective PA interventions have too often been conducted only in controlled settings and more efforts are required to implement and evaluate interventions in real-world contexts [44]. Although the number of school-based PA interventions that have used the RE-AIM framework for evaluation purposes have increased in recent years [45], to the best of our knowledge, no school-based interventions have used this framework to promote ACS. Evaluating all five components of the RE-AIM framework may improve the translatability, scalability, feasibility, and sustainability of interventions to promote cycling to school.

### 1.3. The Pedalea y Anda al COle (PACO) Study Framework

The “Pedalea y Anda al COle: PACO” [Cycling and Walk to School] study has been designed to respond to the main needs, challenges, and limitations of ACS interventions identified in the literature to date [22,23,24,25,29,31,46]. Currently, there are a limited number of ACS interventions focused on promoting adolescent´s cycling to school. Behaviour change frameworks such as the SDT and SEM have been used as theoretical frameworks to guide the design of the PACO study cycling intervention. Finally, the RE-AIM framework will be used to evaluate aspects related to the internal and external validity of this school-based ACS intervention.

The most successful and effective PA interventions in real-world settings are those that have utilized a theoretical framework, identifying the intervention effects on potential mediators [47]. Grounded in SDT and SEM frameworks, as well as the existing evidence about potential mediators of PA [48,49,50], several mediators have been identified in the PACO study to determine which components of an intervention contribute to behaviour change.

The PACO study is a randomized controlled trial study with three main aims developed according to a logic model that identifies mediators and outcomes (see Figure 1).

## 2. Methods

### 2.1. Study Design

The PACO study is a randomized controlled trial setting in public high schools from three Spanish cities (Granada, Jaén, and Valencia), where the schools are the units of randomization, and individuals within the schools are the units of analysis (the participants). The intervention focuses on promoting cycling to school and will be implemented in adolescents during 1 month within the physical education (PE) curriculum classes. Six high schools (i.e., an intervention and a control school in each of the 3 involved cities) will be randomly selected to participate. All study participants will be assessed before and after the intervention period using assessment procedures described below.

The PACO study has been approved by the Review Committee for Research Involving Human Subjects at the University of Granada (Reference: 162/CEIH/2016) and registered with a ClinicalTrials.gov ID: NCT03937336. In addition, the present study protocol has been written in accordance with the Standard Protocol Items: Recommendations for Interventional Trials (SPIRIT) statement (see Appendix A).

### 2.2. Study Population

A total of 360 adolescents aged 14–15 years (3rd grade of secondary education; hereinafter called 3rd grade) from 6 high schools located in 3 Spanish cities will be recruited (see Figure 2). The randomization process is presented in the Figure 2. Within each city, 2 high schools, randomly selected and allocated to intervention or control treatment, will participate in the study. Within each secondary school, two class groups of approximately 30 participants each will be invited to participate and allocated to intervention or control conditions. The final target sample will comprise 360 adolescents (120 adolescents per city; 180 adolescents in the intervention and 180 in the control group).

### 2.3. Procedure

#### 2.3.1. School Recruitment and Randomization

A list of all public high schools from each of the participating cities will be obtained from the educational government institutions (see Figure 3). The schools will be randomly selected in each city separately and a final ordered list will be created. The inclusion criteria for schools´ participation will be: (1) the school must have at least 2 class groups of 3rd grade of secondary education, (2) each class group must include at least 15 adolescents, (3) there must be at least 15 informed consents per class group, (4) the adolescents must have not taken part in other interventions to promote ACS or PA during the PACO study intervention period, and (5) the school must not offer bus transportation to the adolescents. If a school decides not to participate in the study, the same procedure will be carried out with the next selected school from the ordered list.

#### 2.3.2. Recruitment of Participants and Informed Consent

In the participating schools, the research staff will recruit the participants visiting the selected classes and explaining the study to prospective adolescent participants. Adolescents who do not participate in the PE lessons due to any physical or mental diseases at the time of data collection would be excluded from the study. Only adolescents with signed parental consent will be eligible to participate.

#### 2.3.3. Sample Size Calculations

A minimum sample size of 300 participants will be needed to detect an odds ratio of 1.5 in the primary outcome (i.e., PA), assuming an alpha error of 0.05 and 80% of statistical power [51]. Therefore, 50 participants regardless of gender per school (6 schools) are needed. To account for the potential loss to follow-up of up to 50% participants regardless of gender (based on investigators’ previous experience due to the use of wearable devices), a total of 360 adolescents will be recruited (i.e., 120 participants per city). The groups randomization will be done at a 1:1 ratio at city level (i.e., from each city, 60 participants will be randomized to intervention and 60 to control groups).

## 3. Description and Rationale of the PACO Study

### 3.1. Pilot Phase of PACO Study

The school-based intervention (including both the intervention content and process evaluation tools) was piloted in 2018. The intervention was initially designed by research staff and subsequently reviewed by research experts and three PE teachers. The intervention was then piloted to ensure its feasibility and was implemented within PE classes in 14 3rd grade adolescents from one school in Granada, Spain. Some improvements, such as the number of activities and the session organization were modified based on observations by research staff, an interview with a PE teacher, and a focus-group with adolescents performed after the pilot intervention.

### 3.2. Description of the Intervention in the PACO Study

The PACO study comprises a school-based cycling intervention that will aim to promote cycling to and from school and to increase PA levels, while addressing relevant factors within the PE curriculum for secondary education. It is based on the intervention strategies proposed by the Active Living by Design Community Action Model framework [52] including: *preparation* through developing an intentional intervention, *promotion* through educating and encouraging participants on adopting cycling behaviours, and *programs* through organizing activities to engage the participants. These three strategies have been the most commonly used in published ACS interventions [22,25].

The research staff designed the intervention based on previous experience with other school-based ACS interventions and evidence on effective school-based strategies to promote ACS [22,25] and, particularly, cycling to school [53,54]. A main priority was to design a feasible school-based intervention to be implemented within the PE sessions. The intervention objectives, content, components (i.e., knowledge, skills, and attitudes), and the methodological approach are based on the current Spanish national educational law for compulsory secondary education [55]. The contents of the four intervention sessions have been published [56] and are available online (http://profith.ugr.es/pages/investigacion/recursos/manualbici/) to be used by any teacher. Consistent with the RE-AIM framework [42], if the intervention proves effective, it might contribute to promote the adoption and further scale up of this intervention.

The PACO study intervention is based in the Bikeability methodology [57], as a cycle training intervention to gain practical skills and understand how to cycle on roads. Given that the intervention objective is to promote cycling to school, the intervention components and contents are focused on: (a) *knowledge* about road safety rules, cycling safety equipment, cycling hand signals, and the benefits of cycling as a mode of commuting; (b) *skills* related to cycling in a traffic-free and on-road in urban context, and (c) *attitudes* related to awareness, confidence, enjoyment, and usefulness of cycling.

The 7 h intervention has four sessions that will be implemented during PE sessions over a period of one month (1 weekly session). An expert teacher in cycling in an urban context will lead all the intervention sessions, and the PE teachers will attend to learn and support the sessions. The sessions are described below (see Figure 4):

(1) Theoretical session (60 min—first week): theoretical session in the classroom to provide the participants a first step in the concept of cycling as a mode of commuting in the city. The content will include: awareness about the benefits and usefulness of cycling as a mode of commuting, knowledge about road safety rules for cycling, knowledge about cycling safety equipment for both the rider and the bicycle, and cycling hand signals in urban context.

(2) Closed circuit session (120 min—second week): practical session on the school grounds to learn or improve the fundamental cycling skills needed to ride the bicycle safely. The participants will practice how to fit the helmet and check the bicycle before cycling, as well as the fundamental cycling skills through 10 activities and a cycling circuit. The contents will include: correct helmet fitting, bicycle safety check before the ride, and the fundamental cycling skills of starting off and pedalling, breaking safely, changing gears, and hand signals to change directions.

(3) Urban circuit session (120 min—third week): practical session in urban context on-road traffic to transfer the previously learnt cycling knowledge and skills. The participants will learn advanced cycling skills on-road and will use the previously learnt cycling skills in new, uncertain, and modifiable on-road situations exposed to urban traffic. The content will include: starting from side of road (curb), stopping on side of road (kerb), overtaking a parked or slower-moving vehicle, lane changing, turning right and left, and crossing a roundabout.

(4) Bicycle´s party (120 min—fourth week): theoretical and practical session on the school grounds to strengthen previous learning. The participants will become teachers of their younger peers from 1st grade of secondary education. The content will include three activities based on knowledge and fundamental cycling skills learnt in the previous sessions about on-road cycling practice: “fine tuning”, “handling” and “driving”. “Fine tuning” activity will include bike fitting, correct helmet fitting and parking, basic bicycle check, and basic rules of cycling and signalling. “Handling” activity will be a basic cycling skills circuit with tasks including looking back, extending the arm for the right/left turn, looking back and extending the arm for the right/left turn, zig-zag and stopping area, with possible variations (e.g., making the circuit with one hand, changing gears, and grabbing the hand of a partner). “Driving” activity will include exercises to develop the main skills required for road traffic: left and right signalling, reduction/stop signalling, obstacle warning and looking back signalling. The activities will be displayed in a circuit and the younger students will perform these consecutively.

Participating schools will provide the required material for the intervention sessions (i.e., signalling objects -cones and hoops- to build the circuits). Each adolescent participant will need to have access to a bicycle and helmet in the second, third, and fourth session and a reflective vest the third session. Based on previous research experience and previous technical reports [16], it may be unrealistic to expect all participants to own and bring bicycles to the secondary school. Consequently, some alternative strategies will be considered for implementation of the PACO study intervention: (a) in the second and fourth session, half of the students will perform the activities first, followed by the other half, (b) since cycle training session schedules are different for each class group, students in one class could lend their bicycles to students in another class, (c) some high schools have their own bicycles that are offered freely to the students, and d) local traffic or commuting organizations can be asked to lend the bicycles for the intervention.

Regarding the methodology, the sessions will promote the student´s participation and the language used for communication will be inclusive and adapted to them. Active participation of the students will be promoted through questions, encouraging dialogue, and providing meaningful individual and group feedback. The urban circuit (third session) will be supported by expert cycling instructors and school teachers, to accomplish the ratio of 2 adults (i.e., one cycling instructor and one schoolteacher) per 10 participants.

## 4. Measurement Procedures and Outcome Measures

This study will gather data using questionnaires, anthropometry measurements, accelerometry, and the Global Positioning System (GPS).

### 4.1. PACO Study Assessment Protocol

In the PACO study, assessments will be performed at baseline (before the 4-week intervention period) and immediately after the intervention period for both intervention and control groups (see Figure 5). A further detailed description has been published [58]. All measurements will be completed at participant’s school by trained research staff during 3 assessment sessions conducted over 9 days both at baseline and after the intervention period:Assessment session 1 (1 h): this session will include anthropometry assessment at an indoor sport facility. During this session, participants will receive their accelerometer and the activity diary as well as a family questionnaire for their parents to complete at home.Assessment session 2 (1 h): participants will receive their GPS unit and corresponding instructions.Assessment session 3 (1 h): during this classroom-based session, participants will return their accelerometer, the activity diary, the GPS unit, and the completed family questionnaire. During this session, participants will also complete a student questionnaire online or on paper. The questionnaires will be subsequently read using DataScan [59] to guarantee an objective process.

### 4.2. Outcome Measures

The primary outcomes are adolescent´s behaviours (i.e., cycling to school, ACS–walking and cycling, and PA). Other variables include a range of individual, interpersonal, and environmental factors that are related to ACS (see Table 1), and they will be potential mediators or covariates in the statistical analysis. All the questionnaires self-reported by students will be compiled in a global Student questionnaire; and all the questionnaires self-reported by families will be compiled in a global Family questionnaire.

#### 4.2.1. Primary Outcomes

(1) Cycling and active commuting to and from school

The Student questionnaire includes the *Mode and Frequency of Commuting To and From School Questionnaire* which has been previously validated with accelerometer in Spanish children and adolescents [60] and its reliability has been recently assessed [61]. It is a 4-item self-report instrument designed to evaluate the usual mode and weekly frequency of mode of commuting in children and adolescents. The four questions included in the questionnaire are: (1) *How do you usually get to school?*; (2) *How do you usually get home from school?*; (3) *How did you get to school each day?*; and (4) *How did you get home from school each day?*; answer options: (a) *walk*; (b) *bicycle*; (c) *car;* (d) *motorcycle*; (e) *bus*; (f) *public bus*; (g) *metro/train/tram*; or (h) *other* (the mode description was required). The final variable to analyse will be the percentage of usual cycling and active modes of commuting to and from school, and the weekly number of cycling and active travels to and from school. Furthermore, two relevant questions closely related to the active commuting behaviour will be completed by the participants: the distance and time to school. The questions are: (1) *How far do you live from school?;* answer options: (a) *<0.5 km;* (b) *0.5 km to <1 km;* (c) *1 km to <2 km;* (d) *2 km to <3 km;* (e) *3 km to <5 km;* and (f) *5 km or more*; and (2) *How long does it take to get to the school since you leave your house?*; answer options: (a) *<15 min;* (b) *15 min to <30 min;* (c) *30 min to <60 min;* and (d) *60 min or more*. Travel time will be classified as “<15 min” if they commuted to school taking less than 15 min and “≥15 min” if they were commuted to school taking 15 min or more [69].

In addition, the parents will complete the family commuting-to-school behaviour questionnaire at home, self-reporting their children´s usual mode of commuting to and from school and the distance and the time to school, using similar questions to those previously reported by the adolescents. A recent study has concluded a very good reliability between children and parents’ self-reports about the mode of commuting to and from school, distance, and time to school in children and adolescents [62].

(2) Physical Activity

PA will be measured objectively with a triaxial accelerometer (Actigraph wGT3X-BT, Pensacola, FL, USA) for 7 consecutive days, during waking time. Adolescents will be instructed to use an accelerometer attached to the waist on the right side of the body. Additionally, they will be instructed in the care of the device and it will be removed during water activities and sleeping time during night. Students will also complete an activity diary to record when the accelerometer is removed, the time when they wake up and go to sleep, as well as, the time when they go out from home to school. The Actilife software (Actigraph, v.6, Pensacola, FL, USA) will be used both for the initial configuration before the evaluations and for the data dump registered during the evaluation week. Accelerometers should be programmed with a frequency of 90 Hz and an epoch length of 15 s according to the recommendations for the sample of this study (adolescents) [70]. The final variables to analyse will be the daily minutes of light, moderate, vigorous, moderate to vigorous, and total PA.

In addition, we will also extract accelerometer data from specific times of the day for each participant, calculating the PA during commuting from home to school, commuting from school to home, in-school and out-of-school hours. Participants will carry a GPS (model Q STARZ BT-Q1000XT -Q STARZ International Co., Ltd. Taipei, Taiwan) on the left side of the body on the same belt of their accelerometer during the last two consecutive week days carrying the accelerometer.

Moreover, the PA will be self-reported through the questionnaire Youth Activity Profile, which provides a simple, low-cost, and educationally sound method that have already been calibrated and validated to accurately estimate children’s MVPA and sedentary behaviour at the group level [71,72]. A Spanish version of the Youth Activity Profile questionnaire was elaborated using a back-translation process and it has been shown to be a feasible and reliable questionnaire in Spanish adolescents [63]. The Youth Activity Profile questionnaire includes a total of 19 items, including 4 general items and 15 specific items divided into three sections: (1) activity at school, (2) activity out of school, and (3) sedentary habits. The final variables to analyse will be an average score ranging from 1 to 5 (lowest to highest) about the PA at school and PA out-of-school.

#### 4.2.2. Other Variables

(1) Sociodemographic characteristics

Participants and their parents will complete questions on sociodemographic characteristics, such as age, school grade and class, gender, full postal address and children´s bicycle ownership.

(2) Cycling knowledge and skills

The knowledge about route safety rules, cycling hand signalling, and circulation will be self-reported by participants at classroom, completing a 20-questions test with multiple-choice answers [56]. A final score will be obtained with the number of correct answers.

The cycling skills of the participants will be measured during the second and third session of the school-based intervention using two ad hoc observational checklists completed by the cycling teacher after every participant´s performance indicating a dichotomic answer (*yes/no*). The observational checklist for the second session on cycling traffic-free includes 18 skills grouped in 6 categories (i.e., starting off, pedalling and changing gears; turning right; turning left; performing zig-zag; breaking; unexpected turning) and the observational checklist for the third session on cycling on-road includes 32 behaviours grouped in 7 items (i.e., starting from side of route (kerb) and starting off; overtaking and lane changing; turning left to a secondary street; turning right to a secondary street; turning left to a main street; turning right to a main street; crossing a roundabout), and both includes a final item about the participant´s general behaviour. A final score will be obtained by every participant of 0–18 points in the first checklist and 0–32 points in the second checklist. For safety reasons, the participants that will not achieve at least 10 points and six compulsory behaviours related to perform safety right and left turns in the first checklist during the traffic-free cycling lesson, will not be invited to participate in the next on-road cycling session. The checklist is also available elsewhere [56].

(3) Perceptions

Participants will complete questions about their perceived barriers to ACS using the previous valid and reliable BATACE (*BArreras en el Transporte Activo al Centro Educativo*) Scale in Spanish adolescents [64]. It is an 18-item scale related to the environment/safety and planning/psychosocial factors. The final variable to analyse will be an average score ranging from 1 to 4 (strongly disagree to strongly agree), for every item and for every factor.

(4) Psychosocial

Participants will complete two questionnaires about the psychosocial SDT-related variables of autonomy, competence, and relatedness satisfaction in ACS and, motivation for ACS.

The basic psychological needs of autonomy, competence, and relatedness satisfaction in ACS will be determinate using the valid and reliable *Basic Psychological Need Satisfaction in Active Commuting to and from School* Scale [65]. It is a Spanish version adapted from the *Basic Psychological Needs in Exercise Scale to PE* [73] to the domain of ACS. The scale consists of 12 items that assess: autonomy satisfaction, competence satisfaction, and relatedness satisfaction in ACS, introduced by the question *“What do you think about your usual mode of commuting to and from school?”* Students’ responses to the scale will be performed on a 5-point scale ranging from 1 to 5 (strongly disagree to strongly agree).

The motivation for ACS will be measured using the valid and reliable *Behavioural Regulation in Active Commuting to and from School (BR-ACS)* questionnaire [66]. It is a Spanish version adapted from the *Behavioural Regulation in Exercise Questionnaire-3* [74] to the behaviour of ACS. This new scale contains 6 factors and 23 items assessing: intrinsic motivation, integrated regulation, identified regulation, introjected regulation, external regulation, and motivation. Responses to the statement “I go or would go to and from school walking or cycling because …” will be recorded on a 5-point Likert-type scale ranging from 0 to 4 (not true for me to very true for me).

(5) Anthropometric

Anthropometric and body composition will be assessed following the recommendations of the International Society for the Advancement of Kinanthropometry [75]. A digital platform and a height meter will be used to assess weight and height. To assess the waist circumference, a non-elastic tape measure will be used. Body mass index will be calculated as the weight in kilograms divided by the square of the height in meters. More information is described elsewhere [58].

(6) Family socioeconomic status

The family socioeconomic status will be self-reported by adolescents using several questions from the Family Affluence Scale II [76] and by parents using several questions from the Family Affluence Scale (FAS) III [67]. In addition, the parents will self-report the familiar income and the parental highest educational level [69].

(7) Parental perceptions

The perceived barriers to allow their children to commute actively to school will be assessed using the previous valid and reliable PABACS (Parental Perception of Barriers towards Active Commuting to School) questionnaire in Spanish children and adolescents [68]. It is a 23-item questionnaire categorized into 3 scales (general, walking, and cycling barriers) designed to determine the main parental barriers. The final variable to analyse will be an average score ranging from 1 to 4 (strongly disagree to strongly agree), for every item and for every scale.

(8) Peer social support

The peer social support to ACS will be self-reported by parents using a question from the School Travel and Safety survey used in the “Walk to School” study among American children [5]. The question will be: *“How often do other people in the neighbourhood walk or cycling with children to/from school?”* Response options: *everyday, a few times a week, a few times a month, a few times a year, or never*. The final variable to analyse will be the percentage of parents at a given school who will respond “everyday” or “a few times a week”.

(9) School characteristics

The student enrolment will be a measure of the school size (i.e., the number of students enrolled in the school during the year of participating in the current study). The school socioeconomic status will be calculated in accordance to previous studies [77,78]. Then, school neighbourhoods will be objectively analysed and classified as high or low socioeconomic status based on a Geographic Information System. The school engagement (to define its implication within the study since it might influence the intervention effects obtained in the study) will be 3 study reporting 3 variables per school: “teacher’s engagement” (i.e., the average of the researcher staff perceptions of the teacher’s engagement from all the classes); “student’s engagement” (i.e., the average of the researcher staff perceptions of the student’s engagement from all the classes); and “school’s engagement” (i.e., the average of the previous teachers and student’s engagement).

(10) City characteristics

The population density (number of inhabitants per city area in km^2^ -inhabitants/km^2^-) will be obtained from the Ministry of Finance and Public Administration of Spain, using the available data closest to the data collection year in each city. The city income from the data collection year will be obtained from the Spanish Public Tax Agency.

(11) Home–school distance

The distance between home and school will be objectively calculated in two ways. A first way previously used [79] will be to calculate the shortest walking network path between the home and the school of each participant using Google Maps™ software. A second way will be obtained combining GPS and Geographical Information System data to establish the real route between home and school. In both cases, the familiar postal address, that will be self-reported by participants and their parents, and the school’s postal address will be used as reference points.

(12) Home and school neighbourhoods

The length of the cycling lane network in a school area within a 5.1 km street-network buffer will be calculated using a Geographical Information System, both for the school´s intervention groups and control groups. The cut point of 5.1 km is stablished considering the threshold distance for cycling to school among Spanish young adults [80]. In addition, the availability of bicycle racks at the schools (or near the main entrance) will be required to the school staff. Cycle lanes data have been obtained through OpenStreetMap (OSM), which data are available for free use under the OSM Open Database License (ODbL).

(13) Weather

Weather variables will be obtained from the National Weather Data Bank, an open source with climatological information recorded by observatories throughout the country. We will include data on temperature (maximum, minimum, and mean), total rainfall, and mean wind speed, collected by the weather station nearest to each school during each of the five weekdays, and the weekend prior to the Student questionnaire completion, in both baseline and follow-up measurements.

## 5. The RE-AIM Framework

The RE-AIM framework provides key indicators for assessing the internal and external validity of the PACO study, which are shown below (see Table 2).

## 6. Data Analysis

Descriptive characteristics of the sample will be reported as means and standard deviations for continuous variables, and frequencies and percentages for categorical variables. Analysis of variance and Chi-square tests will be conducted to determine differences in all variables between study group (i.e., control or intervention) at baseline.

Firstly, the appropriateness of the randomization process will be analysed by exploring outlier values and the normal distribution of the variables using the Kolgomorov–Smirnov test and graphical procedures (normal distribution plots). After inspecting outliers and extreme values, these will be winsorized using the 1st percentile and 99th percentile of the distribution of variables.

Secondly, analysis of covariance (ANCOVA) will be used to analyse the main effects of the study group (fixed factor) on cycling to school, ACS, PA levels, SDT-related variables in ACS, and individual, interpersonal, and environment variables related to ACS from the SEM (dependent variables). The effect size between groups will be calculated according to Lipsey and Wilson (2001), with the following formula: r = [t^2^/(t^2^ + gl)], where t = value of the Student t statistic and gl = degrees of freedom. Additionally, to examine change between baseline and post-intervention, regression models will be used, including the study group in the model as fixed factor and the mean difference (post-intervention minus baseline) of outcome variables as dependent variable.

Finally, to analyse the influence of covariates in the association of the intervention and the main outcome (i.e., cycling to school, ACS, and PA levels), mediation analyses will be performed using the PROCESS macro for SPSS (SPSS Inc., Chicago, IL, USA). For these analyses, the intervention will be included as independent variable and main outcomes as dependent (i.e., cycling to school, ACS and PA levels). Mediators that will be analysed are cycling knowledge, cycling skills, basic psychological needs satisfaction in ACS, motivation for ACS, barriers to ACS from adolescents and parents, and peer social support for ACS.

The following covariables will be included in the regression models: children´s age and gender, date of birth, school grade and name, class, bicycle ownership, parent´s age and gender, weight, height, waist circumference, body mass index, family income, parental educational level, student enrolment, school socioeconomic status, school engagement, population density, city income, distance home-school, cycling lane length, bicycle racks and weather.

Considering that our main aim is to assess real-world effectiveness of the intervention, we will conduct a primary analysis based on an intention-to-treat principle (in which we will analyse participants according to the groups originally assigned). Then, we will conduct a sensitivity analysis per protocol (in which we will exclude participants who did not complete the intervention and/or did not reach a minimum of 75% of attendance).

Analyses will be conducted using the Statistical Package for Social Sciences (SPSS, v. 25.0, IBM SPSS Statistics, IBM Corporation), and the level of significance will be set at *p* < 0.05.

## 7. Discussion

The PACO study will examine the effects of a school-based cycling intervention on adolescent´s cycling to school, ACS, and PA. Several possible mediators influencing the effects of this intervention will be examined based on SDT and SEM frameworks. In addition, an evaluation of its potential for public health impact using the RE-AIM framework will be conducted.

The main purpose of the PACO study has clear social implications, since the school-based intervention has been designed and piloted prioritizing feasibility and usefulness for implementation during PE sessions in secondary schools. The PACO study has been recreated to fit the PE and school context, to ensure a more likely successful implementation [81]. The difficulty of implementing quasi-experimental and randomized controlled trial designs in the school setting due to high variability within schools in relation to factors such as teachers, students, and organizational issues is well known [46]. Previous reviews on school-based interventions to promote ACS outlined the low quality of intervention designs used [22,25], identifying only few RCTs and a lack of theoretical frameworks. In this regard, the PACO study has combined two approaches to be relevant to both research and practice. The randomized controlled trial design used in this study is one of the major strengths. In addition, the study incorporates several theoretical frameworks. The study also provides user-friendly material for promoting cycling in schools that may contribute to the maintenance and sustainability of the intervention in the medium and long terms.

In the search for balance, several pragmatic adaptations will be made to conduct the research in “real-world” scenarios (such as school) where decisions need to be made on how to best conduct the study with the limited amount of time and resources that researchers may have [82]. In the PACO study, these adaptations pertain to the study design, implementation, and evaluation. Firstly, the school teachers will be able to choose the participating class groups, within the same grade. The pilot study and several consultations with school teachers showed that teachers want to choose the participating class and this will increase their involvement in the intervention. Secondly, all high schools in the three participating cities will be included in the random selection processes and then, the chosen schools will be invited to participate. Time limitations do not allow to visit each school for this purpose, and emails to schools have been previously sent (in previous studies of the research group) and they have not been answered because of high workload. Consequently, we will include all the schools in the random process to guarantee that everyone has the same likelihood to be selected. Thirdly, the intervention includes 4 sessions during a 1-month period. Research staff will offer the option to teachers of spreading the 2 h lesson of the 2nd and 4th session in two 1 h sessions using the normal PE schedule. In Spain, the national educational system in secondary schools mandates 2 h per week of PE. Fourthly, regarding the organization of the academic year in 3 trimesters, the study will be implemented in full trimesters to avoid holiday times, including the assessments. Finally, the cycling skills will be assessed by the teacher using a checklist twice during the intervention, simulating real evaluation conditions. However, safety (e.g., to avoid risky activities where children perform skills that they have not learned yet) and time limitations (e.g., lack of more PE sessions to enlarge the intervention), will prevent the cycling skills assessment being carried out at baseline and follow-up, respectively. To summarize, doing research in schools is important to prevent unhealthy behaviours and related beliefs common in our society. Achieving a balance between the scientific requirements of randomized designs and the constraints imposed by real world circumstances is necessary to improve the quality of research conducted in educational settings [83]. It is also consistent with recent calls to achieve a balance between the established evidence-based pathway and a practice-based evidence pathway, and for a more nuanced approach to appraising the utility of diverse types of evidence [84].

The intended outcomes in the PACO study are to increase the number of participants using cycling or other active modes of commuting to school and PA levels. However, there is evidence about the difficulty of changing behaviours though school-based interventions. For example, the rigorously developed school-based Go Active intervention was no more effective than standard school practice at increasing adolescent PA [83]. Therefore, mediators must be measured to identify the pathways for achieving future behaviour changes. Actually, previous reviews of school-based interventions to promote ACS concluded about the lack of inclusion of mediators in the statistical analyses [22,25]. In the current study, an in-depth analysis of mediators is proposed based on SDT and SEM. The effects of the intervention in the mediators, as well as possible effects of the mediators on the main outcomes, will be analysed. The presence of an appropriate control group is crucial to assess the effects of the intervention. The findings from the PACO study will extend current knowledge about the effectiveness of a cycling school-based interventions to increase active modes of commuting and physical activity levels among secondary school students.

Strengths of the PACO study include the study design (i.e., randomised controlled trial), the inclusion of proposed mediators of the effects of the intervention, the use of theoretical frameworks (i.e., SDT and SEM) and the use of internal and external validity indicators of study design and assessment of implementation processes guided by the RE-AIM framework. The limitations include the low sample size, the short duration of the intervention, the lack of family involvement in the intervention, uncertainties surrounding the engagement and cooperation of school teachers, and the lack of assessment of cycling skills and the maturity at baseline and post-intervention.

## 8. Conclusions

The PACO study will implement a school-based intervention to promote cycling. It will yield a comprehensive understanding of the effects of the intervention on cycling to school behaviour, ACS, PA, basic psychological needs satisfaction in ACS, motivation for ACS, adolescent´s and parental perceived barriers to ACS and peer social support for ACS in Spanish adolescents. Moreover, the PACO study will examine the mediator effect of the previous variables in the adolescent´s cycling and ACS behaviours and overall PA. This intervention will contribute to promote a healthy active commuting behaviour among adolescents to create more liveable and sustainable cities. The PACO study findings have potential to provide guidance for teachers, researchers, and policy makers to implement effective interventions to promote cycling to school and contribute to healthier societies.

## Figures and Tables

**Figure 1 ijerph-18-02066-f001:**
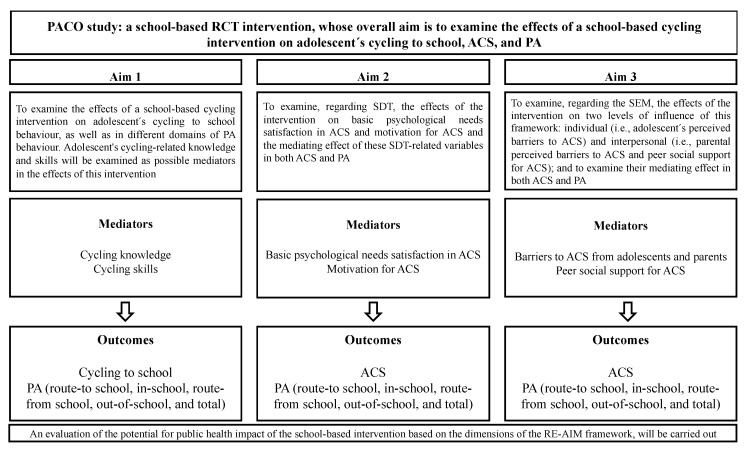
The “Pedalea y Anda al COle” (PACO) study logic model. Note: RCT: randomized controlled trial; ACS: active commuting to and from school; PA: physical activity; SDT: self-determination theory; SEM: social-ecological model; RE-AIM: reach, effectiveness, adoption, implementation, maintenance.

**Figure 2 ijerph-18-02066-f002:**
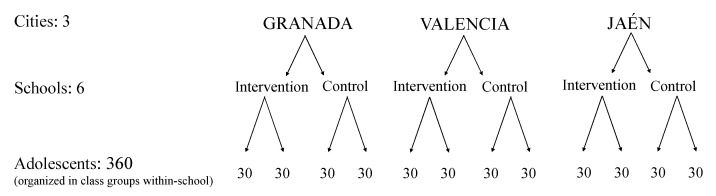
Cities, schools, and adolescent participants in the PACO study.

**Figure 3 ijerph-18-02066-f003:**
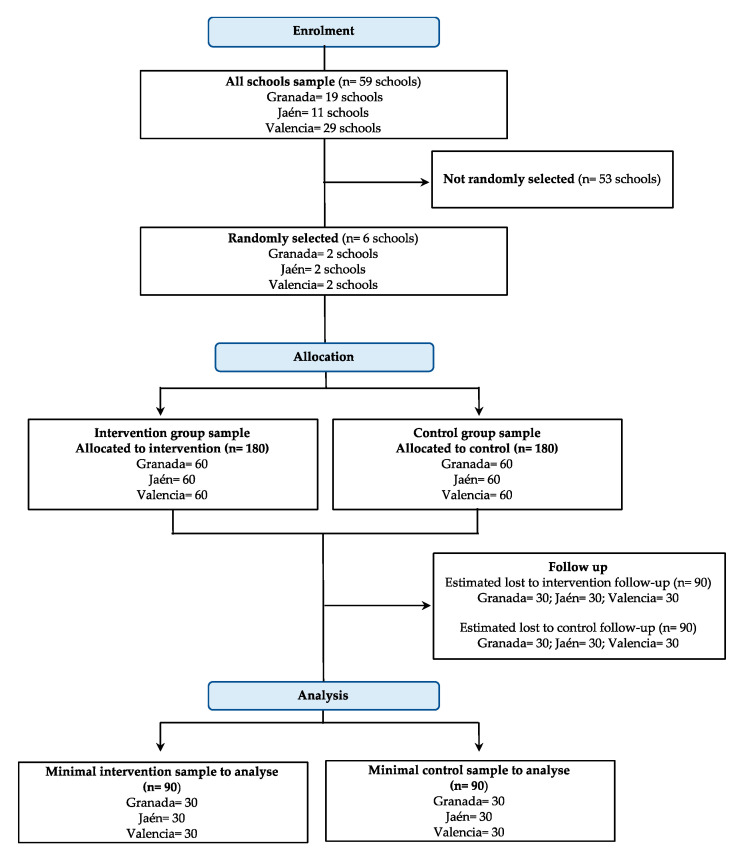
Plan flow chart of the PACO study.

**Figure 4 ijerph-18-02066-f004:**
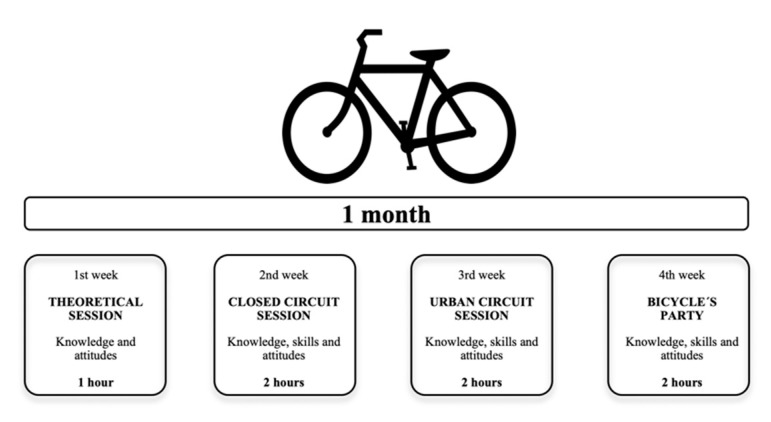
Structure and duration of the school-based intervention to promote cycling to school.

**Figure 5 ijerph-18-02066-f005:**
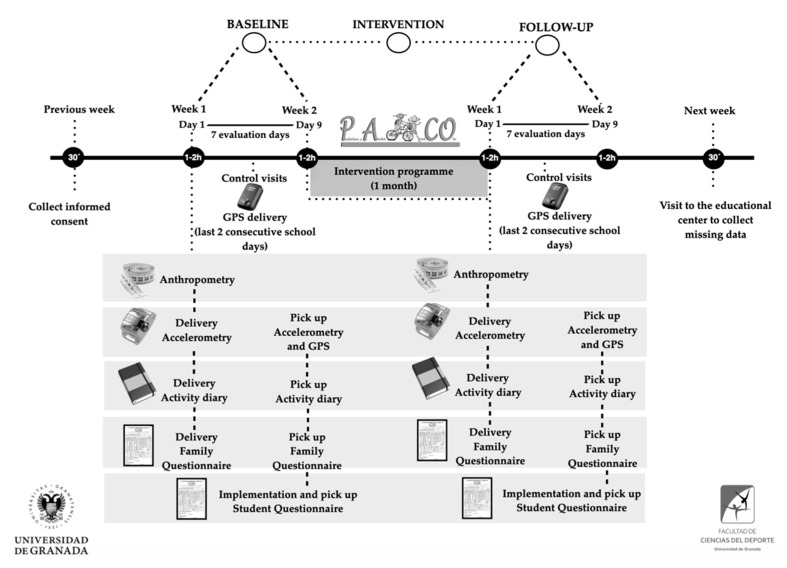
The measurement procedure of the PACO study.

**Table 1 ijerph-18-02066-t001:** Summary of the variables and instruments, following the levels of influence of the social-ecological model (SEM).

General Variable	Specific Variables	Instrument
**Individual**		
Sociodemographic characteristics	Age, school grade and class, gender, full postal address, bicycle ownership	Student questionnaire
Behaviour	Cycling and active commuting to/from school (ACS) Physical activity (PA) Cycling knowledge, cycling skills	Mode and Frequency of Commuting to/from School questionnaire [60,61] Family commuting-to-school behaviour questionnaire [62] Accelerometer Youth Activity Profile questionnaire [63] Cycling Tests
Perceptions	Perceived barriers to active commute to school	Barreras en el Transporte Activo al Centro Educativo (BATACE) questionnaire [64]
Psychosocial	Autonomy, competence, and relatedness satisfaction in ACS Motivation for ACS	Basic Psychological Need Satisfaction in Active Commuting to and from School questionnaire [65] Behavioural Regulation in Active Commuting to and from School questionnaire [66]
Anthropometric	Weight, height, waist circumference, body mass index	Weighting platform Seca 876 Height meter Seca 2013 Non-elastic tape measure Lufkin W606pm
**Interpersonal**		
Parental sociodemographic data	School name, age, gender, children´s gender, and full postal address	Family questionnaire
Family socioeconomic status	Family income Parental educational level	Family Affluence Scale (FAS) questionnaire [67] Family questionnaire
Parental perceptions	Barriers to allow their children to ACS	Parental Perception of Barriers towards Active Commuting to School (PABACS) questionnaire [68]
Peer social support	Peer social support	Family questionnaire
**Community**		
School Characteristics	Student enrolment, socioeconomic status and school engagement	Electronic and manual Search, Geographical Information System,
City Characteristics	Population density and city income	Tax Agency, Spanish Public Ministry of Finance and Public Administration of Spain
**Environment**		
Distance home-school	Shortest route, real route	Google Maps Global Positioning System (GPS)
School Neighbourhood	Cycling lane length Bicycle racks	Geographical Information System Manual Search
Weather	Temperature (maximum, minimum, mean), total rainfall, and mean wind speed	National Weather Data Bank

**Table 2 ijerph-18-02066-t002:** Reach Effectiveness Adoption Implementation Maintenance (RE-AIM) components and dimensions to be evaluated in the PACO study.

RE-AIM Component	PACO Study
REACH (R)	The absolute number and proportion of participants recruited for the study given a target sample of 360 adolescents aged 14–15 years old from the 3rd grade of secondary education.
EFFECTIVENESS (E)	Main outcomes: cycling to school, ACS—walking and cycling, and PA Other variables: cycling knowledge, cycling skills, basic psychological needs satisfaction in ACS, motivation for ACS, barriers to ACS from adolescents and parents, and peer social support for ACS.
ADOPTION (A)	The absolute number and proportion of school staff who are willing to deliver the intervention. Alignment/consistency between intervention content and national educational policy [55].
IMPLEMENTATION (I)	Intervention fidelity: extent to which intervention agents deliver the intervention components as intended, including adaptations made, and of participant compliance with the intervention in terms of attendance. Cost of the intervention Process evaluation: participant feedback about enjoyment, usefulness, and potential improvements after each session. Participant focus groups and interviews with intervention agents at the end of the intervention to understand their experiences of the intervention.
MAINTENANCE (M)	Setting level: extent to which the intervention is delivered as part of the school PE curricula the following academic year. Individual level: follow-up questions to the PE teachers after 6 months post-intervention to assess the intervention´s maintenance.

## Data Availability

Not Applicable.

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
