# Peer review of "A School-Based Randomized Controlled Trial to Promote Cycling to School in Adolescents: The PACO Study"

_ijerph, 2021, doi:10.3390/ijerph18042066_

Round 1

Reviewer 1 Report

I believe the authors tap into an important topic in sport science and health in promoting active commuting to school in adolescents.

This study protocol aiming to describe “A school-based randomized controlled trial to promote cycling to  school in adolescents: The PACO study”

This paper explains in a positive way when looking at the topic of adolescents health and strategies to improve their daily physical activity. I think some things need clarifying for the publication that will help in the overall interpretation and understanding of the futures results.

Comment 1: Line 87 – “ A recent systematic review….” - Please add the reference of the systematic review.

Comment 2: Line 188 – “2.3.2 Recruitment of participants and informed consents” – Can the authors add criteria inclusion and exclusion for this study (i.e: criteria exclusion: cardiovascular,  respiratory or neurologic diseases)? All adolescents can be participating in this study?

Comment 3: Line 192 – “ 2.3.3 Sample size calculations” - The number of samples was calculated for boys and girls? The authors considered a minimum of boys and girls for each group? The authors considered 50% for each sex? Can the authors specify this calculation?

Comment 4: Line 256 – “Urban circuit session” - Based on geographic characteristics, Granada, Jaén and Valencia are different cities. Granada irregular (many climbs) and Valencia is majority plane. These characteristics can promote different grades of difficulty (intensity and technical). The weather is an important variable for promoting ACS. Also, the winter temperature is not the same in Granada and Valencia. The authors will realize the study at the same season of the year for all cities? How the authors will control these factors?

Comment 5: Line 297 – “ 4. Measurement procedures and outcomes measures – This study will gather data using questionnaires, anthropometry measurements…” -  The maturity is an important factor in the sample of this range age (adolescent). Why the authors did not consider this variable?

I think the manuscript should be improved. I have just some minor comments that I wish the authors and editor take into account before to publish.

Author Response

Comment 1

  • In I believe the authors tap into an important topic in sport science and health in promoting active commuting to school in adolescents. This study protocol aiming to describe “A school-based randomized controlled trial to promote cycling to  school in adolescents: The PACO study”. This paper explains in a positive way when looking at the topic of adolescents health and strategies to improve their daily physical activity. I think some things need clarifying for the publication that will help in the overall interpretation and understanding of the futures results.

Answer 1

  • Thank you very much for your helpful comments and suggestions. We have addressed all your comments below. Moreover, we have incorporated them into the revised manuscript, and we believe our manuscript is stronger as a result of these modifications. All the changes to the original manuscript are highlighted in yellow.

Comment 2

  • Line 87 – “ A recent systematic review….” - Please add the reference of the systematic review.

Answer 2

  • Thank you for your comment. We have added the reference of the scoping review: Sersli S, Scott N, Winters M. Changes in bicycling frequency in children and adults after bicycle skills training: A scoping review. Transp Res Part A Policy Pract 2019, 123, 170-187.

Comment 3

  • Line 188 – “2.3.2 Recruitment of participants and informed consents” – Can the authors add criteria inclusion and exclusion for this study (i.e: criteria exclusion: cardiovascular,  respiratory or neurologic diseases)? All adolescents can be participating in this study?

Answer 3

  • Thank you for your consideration. For this study, we had criteria inclusion for school´s participation (see lines 180-185). Regarding your question, we have added the following information: “Adolescents who do not participate in the PE lessons due to any physical or mental diseases at the time of data collection would be excluded from the study.” (see lines 192-194). Only adolescents with signed parental will be eligible to participate in this study.”

Comment 4

  • Line 192 – “ 2.3.3 Sample size calculations” - The number of samples was calculated for boys and girls? The authors considered a minimum of boys and girls for each group? The authors considered 50% for each sex? Can the authors specify this calculation?

Answer 4

  • Thank you for your comment. The sample size was calculated for number of adolescents per schools (without accounting for gender differences) because our study is not examining gender differences. To clarify your suggestion, we have included “50 participants regardless of gender per school” and “50% participants regardless of gender” (see lines 199-201).

Comment 5

  • This Line 256 – “Urban circuit session” - Based on geographic characteristics, Granada, Jaén and Valencia are different cities. Granada irregular (many climbs) and Valencia is majority plane. These characteristics can promote different grades of difficulty (intensity and technical). The weather is an important variable for promoting ACS. Also, the winter temperature is not the same in Granada and Valencia. The authors will realize the study at the same season of the year for all cities? How the authors will control these factors?

Answer 5

  • Thank you for the suggestion. The study will not be executed at the same season of the year for all cities because it is very difficult to coincide with the school calendar and we do not have enough instruments (i.e., accelerometer, GPS devices) to conduct data collection in all the participants at the same time. In addition, the data collection dates will depend on the organization of the each individual school. However, we are aware of the seasonality, location and weather variables. We have taken those factors in account by measuring relevant variables and including them in statistical analyses (i.e weather, lines 498-504; home and school neighbourhoods lines 490-497).

Comment 6

  • Line 297 – “ 4. Measurement procedures and outcomes measures – This study will gather data using questionnaires, anthropometry measurements…” -  The maturity is an important factor in the sample of this range age (adolescent). Why the authors did not consider this variable?

Answer 6

  • We totally agree with the reviewer. In adolescents, the maturity is an important factor but, in our study, the maturity was not measured. We have been included this aspect on limitations “The maturity was not measured” (see line 629).

Reviewer 2 Report

The protocol is well detailed and identifies the policy and academic gap. However it can be improved by addressing the following gaps/ concerns

  1. The authors should consider minimising repetitions especially in sections 2.1, 2.3 and 3.2
  2. Is it correct to refer to column  2 of table 1 as outcomes?
  3. Please provide the rationale for the choice of the 3 sites and not any other
  4. Aim 3 seem be compound. You could break it in to two
  5. 2.1 need to be summarized into tabular form ( i.e. use of codes)  

Further please attend to the following issues

  1. What are the stopping rules for individuals/part of study/procedures  and conditions of breaking codes
  2. What are the safety consideration- procedures for  recording and reporting  adverse events and their follow-up ( how and for how long)
  3. Level of significance to be used statistical analysis?
  4. What will be the procedure for accounting for any missing or spurious data
  5. What are the anticipated problems, roles and responsibilities
  6. How does the protocol contribute to advancement of knowledge and will this knowledge be utilised .i.e. publications, health care/ systems and health policies
  7. How will dissemination of the results be handled ( In particular the publication policy to be adopted   with respect to  community of practice, community, policy makers and who will  take the lead)

Author Response

Comment 1

  • The protocol is well detailed and identifies the policy and academic gap. However it can be improved by addressing the following gaps/ concerns

Answer 1

  • Thank you very much for your helpful comments and suggestions. We have addressed all your comments below. Moreover, we have incorporated them into the revised manuscript, and we believe our manuscript is stronger as a result of these modifications. All the changes to the original manuscript are highlighted in yellow.

Comment 2

  • The authors should consider minimising repetitions especially in sections 2.1, 2.3 and 3.2

Answer 2

  • Thanks for the appreciation because we have eliminated some repetitions after have carefully reviewing these sections.

Comment 3

  • Is it correct to refer to column  2 of table 1 as outcomes?

Answer 3

  • Thank you for your good suggestion. We have thought about your comments to find another term to the column 2. We have changed “outcomes” by “variables” because we think it is more coherent (see table 1).

Comment 4

  • Please provide the rationale for the choice of the 3 sites and not any other

Answer 4

  • Thank you for your comment. The three cities were chosen based on the feasibility to conduct data collection for this project. We wanted to focus on large cities, and we considered 3 sites cities which have universities with researcher teams that were interested able to implement this project.

Comment 5

  • Aim 3 seem be compound. You could break it in to two

Answer 5

  • We agree withyou’re the reviwers’ suggestion. It is a good idea what the reviewer says to break the objective 3 into two separating the individual and interpersonal level outcomes, since there are more than 1 objective within it. However, to make clear that we are using two frameworks behind the objectives, the Self-Determination Theory and the Socio-Ecological Model, we believe it is better to set all the objectives referring to these frameworks together in the same objective 2 and 3 respectively.

Comment 6

  • 1 need to be summarized into tabular form ( i.e. use of codes) 

Answer 6

  • Thanks for the comments. We have clearly separated the 2.1. section in two paragraphs using tabulations.If we did not understand appropriately the comment, we suggested the reviewer to explain again, please.

Comment 7

  • What are the stopping rules for individuals/part of study/procedures  and conditions of breaking codes

Answer 7

  • We decided to separate the study/procedure in several sections following the proposed temporal order of data collection for this study, to make the readers and other researchers easier the understanding and the potential replication of similar studies. A detailed explanation through different sections may contribute to it. If we did not understand appropriately the comment, we suggested the reviewer to explain again, please.

Comment 8

  • What are the safety consideration- procedures for  recording and reporting  adverse events and their follow-up ( how and for how long)

Answer 8

  • Thank you for your comment. The safety limitation included in the discussion, has been rewriten for a better understanding and clarification.

In addition, we want to clarify the reviewer his/her comment. The safety limitation refers that the cycling skills assessment was not performed at baseline because participants maybe novel and the cycling assessment might provoke accidents. On the other hand, the cycling assessment was not performed in the follow-up neither, due to time limitation for the intervention in the schools (see lines 592-595).

Comment 9

  • Level of significance to be used statistical analysis?

Answer 9

  • Thank you for your comment. At the end of the statistical analysis part, we indicate that: “… and the level of significance will be set at p < 0.05” (see lines 549-550).

Comment 10

  • What will be the procedure for accounting for any missing or spurious data

Answer 10

  • Thank you for your comment. Participants with missing data for specific variable will be excluded in the statistical analysis. We thought on including imputation analysis, but regarding the low sample size and after consulting with statisticians, the decision was not to complete missing data.

Comment 11

  • What are the anticipated problems, roles and responsibilities

Answer 11

  • Thank you for this good suggestion. The potential anticipated problems may be collecting the school and participants consent and suffering an accident during the cycling intervention.Both the research team and the school board will be responsible respectively.

 Comment 12

  • How does the protocol contribute to advancement of knowledge and will this knowledge be utilised .i.e. publications, health care/ systems and health policies

Answer 12

  • Thank you for the relevant comment. We hope that this protocol will be useful for other researchers and practitioners. Firstly, after it is published, we will disseminate it through social networking and international researchers interested on this topic, since we are aware that the current manuscript covers several gaps and provide a new and real advancement. This study protocol will not be disseminate to the health and educational policy since there are not results and it does not show evidence about its effectiveness.

Comment 13

  • How will dissemination of the results be handled ( In particular the publication policy to be adopted   with respect to  community of practice, community, policy makers and who will  take the lead)

Answer 13

  • After the study is implemented, we will analyze the results. The results will be disseminated with scientific andhigh-quality indexed publications and oral and poster presentations in national and international conferences. In addition, conferences in school and community settings will be offered to present the results for the all society.

Reviewer 3 Report

This paper is essentially a study design. The authors have identified a gap in research understanding in relation to cycling to school and have prepared a well crafted design to explore this issue. A more extensive literature review would have been worthwhile. However, the critical weakness of  the paper is that it has no results. The field work remains to be done. The paper should not be considered until it has some results to report. The description of proposed methodologies is not worthy of publication in its own right.

Author Response

Comment 1

  • This paper is essentially a study design. The authors have identified a gap in research understanding in relation to cycling to school and have prepared a well crafted design to explore this issue. A more extensive literature review would have been worthwhile. However, the critical weakness of  the paper is that it has no results. The field work remains to be done. The paper should not be considered until it has some results to report. The description of proposed methodologies is not worthy of publication in its own right.

Answer 1

  • Thank you for your suggestion. As the reviewer stated, we have identified a gap in relation to cycling to school and designed a research study to address that knowledge gap. Please note that the current manuscript describes the study protocol of this comprehensive project . Study results will be published separately in future after data collection is completed in future.

The published scientific literature includes many study protocols and the origin comes from the clinical investigations.

Selected journals, including the IJEPHR, also accept submission of study protocol manuscripts. Please refer to IJERPH website:

www.mdpi.com/search?journal=ijerph&article_type=study-protocol

Reviewer 4 Report

Dear authors,

Thank you for the opportunity to review your manuscript. I find your study protocol to be very promising and I appreciate your approach expertly combining scientific and practical/implementational views. I am sure that your trial has potential to bring many benefits.

The rationale and the protocol of the randomized controlled trial are described in detail and I certainly recommend publication of this manuscript in the journal.

I have only several minor comments (some of them trivial):

- It is recommended not to start a sentence in an abstract with a number. One sentence in your abstract starts with “. 360 adolescents”.
- In the Figure 1, some abbreviations (SDT, SEM, RE-AIM) are not defined in the note. It is good to have figures self-explanatory.
- What is the current status of data collection for the study? Is it planned, ongoing or finished? Isn’t it too late for publication of the study protocol now? I am asking because you mention that the study has been approved by the Review Committee already in 2016 and pilot done in 2018. Please make it clear.
- Is there any specific reason why Granada, Valencia, and Jaén were selected? I am thinking of some variables related to SEM (environmental attributes) that may influence your findings. Did you consider the elevation profile of the cities/school neighborhoods? “Flat/hilly” city/neighborhood may play substantial role in adopting cycling. You may also control for this variable in your models.
- Line 241, typo: “…will lead the all intervention sessions…” -> “…will lead all the intervention sessions…”
- Figure 4 seems to have low information value. Consider omitting it.
- In the Figure 5, at baseline, labels Week 1 and Week 2 may be better (rather than Week 2 and 3).
- Table 1 -> General outcome -> Interpersonal: I find the item “parental behavior” little bit confusing. I am not sure if it is related to parents reporting the behavior of their children, parents actively commuting to schools together with their children or indirect influence of parents who are active commuters and may thus better understand and support active commuting in their children. It is difficult to understand this item in the table without reading the related text.
- Lines 432-433: You mention “…7-point Likert-type scale ranging from 0 to 4…”. It should be probably 5-point scale, however.
- Lines 435-439: Past tense is used in this paragraph. Future tense (as used in other related paragraphs) would be better.
- I am aware that you are addressing it in your study design but there may be two important risk factors: 1) availability of bicycles for the students and 2) availability of bicycle racks at the schools. I don’t know how common it is for the schools in Spain to provide the students with possibilities of safe storing of their bicycles. In this context, I am not sure what this sentence should mean (lines 492-494): “...the presence of bicycle racks inside the school… will be required to the school staff.”
- Lines 565-566: A verb is probably missing in the sentence: “The randomized controlled trial design used in this study one of the major strengths”.

Author Response

Comment 1

  • Thank you for the opportunity to review your manuscript. I find your study protocol to be very promising and I appreciate your approach expertly combining scientific and practical/implementational views. I am sure that your trial has potential to bring many benefits. The rationale and the protocol of the randomized controlled trial are described in detail and I certainly recommend publication of this manuscript in the journal.

Answer 1

  • Thank you very much for your positive comments. We have addressed all your comments below and incorporated them into the revised manuscript. We believe our manuscript is stronger as a result of these modifications. All the changes to the original manuscript are highlighted in yellow.

Comment 2

  • It is recommended not to start a sentence in an abstract with a number. One sentence in your abstract starts with “. 360 adolescents”.

Answer 2

  • We agree with your comment. We have restructured this sentence “A total of 360 adolescents…” (see line 34)

Comment 3

  • In the Figure 1, some abbreviations (SDT, SEM, RE-AIM) are not defined in the note. It is good to have figures self-explanatory.

Answer 3

  • Thank you for your suggestion. We have defined these abbreviations in the note (see lines 147-149).

Comment 4

  • What is the current status of data collection for the study? Is it planned, ongoing or finished? Isn’t it too late for publication of the study protocol now? I am asking because you mention that the study has been approved by the Review Committee already in 2016 and pilot done in 2018. Please make it clear.

Answer 4

  • The current status of the data collection is happening now (2020-2021). In Spain, when you apply for a research grant, it is more valuable if you include the Review Committee. So, we required it in 2016 and then, we requested the project to the Ministry of Spain and we received a positive answer at the end of 2016 and it started in 2017. This big project has 3 main objectives, and the current study protocol (i.e. the intervention study) corresponds to the 3rd part, which is the more novel and challenge part. So, we carried out the 1st and 2nd objectives along 2017-2019, and the 3rd objective (i.e, the intervention) is being carried now.

Comment 5

  • Is there any specific reason why Granada, Valencia, and Jaén were selected? I am thinking of some variables related to SEM (environmental attributes) that may influence your findings. Did you consider the elevation profile of the cities/school neighborhoods? “Flat/hilly” city/neighborhood may play substantial role in adopting cycling. You may also control for this variable in your models.

Answer 5

  • Thank you for your comment. The study will not be executed at the same season of the year for all cities because it is very difficult to coincide with the school calendar and we do not have enough instruments (i.e., accelerometer, GPS devices) to conduct data collection in all the participants at the same time. In addition, the data collection dates will depend on the organization of the each individual school. However, we are aware of the seasonality, location and weather variables. We have taken those factors in account by measuring relevant variables and including them in statistical analyses (i.e weather, lines 498-504; home and school neighbourhoods lines 490-497).

Comment 6

  • Line 241, typo: “…will lead the all intervention sessions…” -> “…will lead all the intervention sessions…”

Answer 6

  • Done

Comment 7

  • Figure 4 seems to have low information value. Consider omitting it.

Answer 7

  • We agree with the reviewer that the Figure 4 is low informative for researching issues and we could omit it. However, we included it thinking that any teacher or school staff –that they are not usually familiarized with the research approach- may read the manuscript and the figure may facilitate to understand the intervention and encourage them to implement it in their lessons. However, if the reviewer consider to omit it, we could do it.

Comment 8

  • In the Figure 5, at baseline, labels Week 1 and Week 2 may be better (rather than Week 2 and 3). 

Answer 8

  • Done

Comment 9

  • Table 1 -> General outcome -> Interpersonal: I find the item “parental behavior” little bit confusing. I am not sure if it is related to parents reporting the behavior of their children, parents actively commuting to schools together with their children or indirect influence of parents who are active commuters and may thus better understand and support active commuting in their children. It is difficult to understand this item in the table without reading the related text.

Answer 9

  • Thank you for catching this mistake. Actually, we have deleted it because we do not measure any parental behaviour. We have rewritten the table 1.

Comment 10

  • Lines 432-433: You mention “…7-point Likert-type scale ranging from 0 to 4…”. It should be probably 5-point scale, however.

Answer 10

  • Thank you for your comment and detecting this typo. We have rewritten it as “…5-point Likert-type scale ranging from 0 to 4…” (see lines 435-436).

Comment 11

  • Lines 435-439: Past tense is used in this paragraph. Future tense (as used in other related paragraphs) would be better.

Answer 11

  • Thank you for this helpful suggestion. We have rewritten this paragraph using future tense.

Comment 12

  • How I am aware that you are addressing it in your study design but there may be two important risk factors: 1) availability of bicycles for the students and 2) availability of bicycle racks at the schools. I don’t know how common it is for the schools in Spain to provide the students with possibilities of safe storing of their bicycles. In this context, I am not sure what this sentence should mean (lines 492-494): “...the presence of bicycle racks inside the school… will be required to the school staff.”

Answer 12

  • Thank you for your comment. In Spain, the availability of bicycle racks at the schools varies by school. For this reason, we believe that it is important to adjust the statistical analyzes taking in account this covariate of “presence or absence of bicycle racks”. However, to collect correctly this data, we will collect that information directly from the school staff from each participating school. Regarding the reviewer comment, we have revised that sentence as follows: “In addition, information about the availability of bicycle racks at the schools (or near the main entrance) will be obtained from the school staff” (see lines 494-496).

Comment 13

  • Lines 565-566: A verb is probably missing in the sentence: “The randomized controlled trial design used in this study one of the major strengths”.

Answer 13

  • Thank you for your comment. We have revised the sentence “The randomized controlled trial design used in this study is one of the major strengths” (see lines 567-568).

Round 2

Reviewer 3 Report

My opinion on this paper has not changed. It should not be considered until it has some results to report.

Author Response

Comment 1

  • My opinion on this paper has not changed. It should not be considered until it has some results to report.

Answer 1

Thank you for your comment. We explained this situation about the publication of protocols in the 1st round of review, so the editor or board members may be able to solve this comment, since the journal IJERPH accepts study protocols (see the webpage: www.mdpi.com/search?journal=ijerph&article_type=study-protocol) and there are a large number of examples. In addition, being consistent with the study protocols, these type of studies have no results section (for a further review see; http://www.prisma-statement.org/Protocols/).

Several study protocols have been published in the journal IJERPH. Please, find below some examples of protocol studies published in the last months:

Macêdo, S. F. D., Silva, K. A., Vasconcelos, R. B. D., Sousa, I. V. D., Mesquita, L. P. S., Barakat, R. D. M., ... & de Oliveira Lima, J. W. (2021). Scaling up of Eco-Bio-Social Strategy to Control Aedes aegypti in Highly Vulnerable Areas in Fortaleza, Brazil: A Cluster, Non-Randomized Controlled Trial Protocol. International Journal of Environmental Research and Public Health18(3), 1278.

Sudeshika, T., Naunton, M., Peterson, G. M., Deeks, L. S., Thomas, J., & Kosari, S. (2021). Evaluation of General Practice Pharmacists: Study Protocol to Assess Interprofessional Collaboration and Team Effectiveness. International Journal of Environmental Research and Public Health18(3), 966.

Armaou, M., Konstantinidis, S., & Blake, H. (2020). The Effectiveness of Digital Interventions for Psychological Well-Being in the Workplace: A Systematic Review Protocol. International Journal of Environmental Research and Public Health, 17(1), 255.

Sooknarine-Rajpatty, J., B Auyeung, A., & Doyle, F. (2020). A Systematic ReviewProtocol of the Barriers to Both Physical Activity and Obesity Counselling in the Secondary Care Setting as Reported by Healthcare Providers. International Journal of Environmental Research and Public Health, 17(4), 1195.

Tracey, D., Gray, T., Sweeting, J., Kingsley, J., Bailey, A., & Pettitt, P. (2020). A Systematic Review Protocol to Identify the Key Benefits and Associated Program Characteristics of Community Gardening for Vulnerable Populations. International Journal of Environmental Research and Public Health, 17(6), 2029.
